# A Protocol for a Systematic Review and Meta-Analysis of Mind–Body Modalities to Manage the Mental Health of Healthcare Workers during the COVID-19 Era

**DOI:** 10.3390/healthcare9101320

**Published:** 2021-10-03

**Authors:** Chan-Young Kwon, Boram Lee

**Affiliations:** 1Department of Oriental Neuropsychiatry, Dong-eui University College of Korean Medicine, Busan 614-714, Korea; 2Department of Clinical Korean Medicine, Graduate School, Kyung Hee University, Seoul 130-701, Korea; qhfka9357@naver.com

**Keywords:** healthcare personnel, mental health, mind–body therapies, COVID-19, pandemics

## Abstract

The coronavirus disease 2019 (COVID-19) pandemic has become an unprecedented threat to humanity worldwide, including healthcare workers (HCWs). Mind–body modalities have been used to improve the mental health, well-being, quality of life, and physical health of clinical and general populations, and may also be used to improve the mental health of HCWs during COVID-19. The objective of this review is to analyze the effectiveness of mind–body modalities for the mental health of HCWs in the COVID-19 era. Six electronic bibliographic databases were comprehensively searched to find intervention studies using mind–body modalities, including meditation, mindfulness-based intervention, autogenic training, yoga, tai chi, qigong, breathing exercise, music therapy, guided imagery, biofeedback, prayer, and faith-based techniques for HCWs. All intervention studies conducted from December 2019 to August 2021 will be included. Quality assessment will be performed according to study type, and Cochrane Collaboration’s Risk of Bias tool will be used for randomized controlled clinical trials (RCTs). If sufficient homogeneous data from RCTs exist, a meta-analysis will be performed. Dichotomous data and continuous data are presented as risk ratios and mean differences with their 95% confidence intervals, respectively. The results of this systematic review will be disseminated through the publication of a manuscript in a peer-reviewed journal or by presentation at a conference.

## 1. Introduction

The coronavirus disease 2019 (COVID-19) pandemic has become an unprecedented threat to humanity worldwide, widely affecting the physical and mental health of people around the world [1]. In particular, healthcare workers (HCWs) have been exposed to the threat of severe acute respiratory syndrome coronavirus 2 (SARS-CoV-2) since the early days of the pandemic [2,3]. In the early stage of the pandemic, the lack of personal protective equipment (PPE), undiscovered etiology, strict isolation, high-intensity work, insufficient rest, fear of infection, and lack of a vaccine were major stressors for them [2,3]. Since then, some vaccines against SARS-CoV-2 have been developed and PPEs have been distributed, but HCWs’ mental health is still threatened by a prolonged COVID-19 outbreak. According to a recent meta-analysis, HCWs exposed to stress from COVID-19 had significantly higher levels of anxiety (13.0 vs. 8.5%) and depression (12.2 vs. 9.5%) compared to experts in other fields [4]. Accordingly, the importance of improving the mental health of HCWs has emerged [5].

A wide range of mind–body modalities designed to facilitate the mind’s capacity to affect health has been used for a long time to improve mental health, well-being, quality of life, and physical health of clinical populations, and the general population in various occupations [6]. Previous studies have reported that mind–body modalities such as yoga [7], mindfulness-based interventions [8,9], and tai chi [10] can benefit the health of HCWs. Today, mind–body medicine has re-emerged for its mental health benefits in the context of COVID-19 [11,12], and some health systems use mind–body modalities to build strategies to improve the mental health of the public, patients with COVID-19, and HCWs [13,14].

Recently, due to COVID-19, some hospitals have reportedly introduced mind–body modalities to improve the mental health of frontline HCWs, including perceived stress [15,16]. Moreover, mind–body modalities are generally considered safe and can also be delivered non-face-to-face in combination with information and communications technology (ICT), making them a promising mental health improvement strategy in the COVID-19 era [17].

However, there has not been a comprehensive review of how mind–body modalities benefit the mental health aspects of HCWs in the context of COVID-19. The objective of this systematic review is to investigate the roles of mind–body modalities in managing the mental health of HCWs during the COVID-19 era. Our findings will help establish better public mental health strategies to improve the mental health of HCWs according to each situation and the available medical resources during this pandemic.

## 2. Materials and Methods

The protocol of this systematic review was registered in the Open Science Framework registry (https://osf.io/tudbw; registered on 13 August 2021). Any amendments to this systematic review will be tracked using this registry. This protocol was reported in accordance with the Preferred Reporting Items for Systematic Review and Meta-Analysis Protocols (PRISMA-P) 2015 statement [18] (Appendix A). We will report our systematic review according to the PRISMA 2020 statement [19].

### 2.1. Data Sources and Search Strategy

One researcher (B.L.) will search the following six electronic bibliographic databases from December 2019 (when the first case of COVID-19 was identified [20]) to September 2021: MEDLINE via PubMed, EMBASE via Elsevier, the Cochrane Central Register of Controlled Trials, Cumulative Index to Nursing and Allied Health Literature via EBSCO, Allied and Complementary Medicine Database via EBSCO, and PsycARTICLES via ProQuest. In addition, we will review the reference lists of the relevant literature and manually search Google Scholar to find potentially missing literature, including gray literature. Since our study will include gray literature and literature published in peer-reviewed journals, we will not evaluate the journal reliability of additional literature identified through manual searches in Google Scholar. However, we will indicate the source of the search for all included articles. The search strategy for Medline via PubMed is presented in Table 1.

### 2.2. Eligibility Criteria

The inclusion criteria for this systematic review are as follows. (1) Types of study: Although randomized controlled clinical trials (RCTs) are a type of clinical study with the highest level of evidence to evaluate the effectiveness and efficacy of the intervention, given the urgency of COVID-19, meaningful results can be found not only in RCTs but also in other intervention studies besides RCT. Therefore, this systematic review will allow for all types of quantitative intervention studies. That is, all original clinical studies, including RCTs, non-randomized controlled clinical trials (CCTs), and before-after studies will be included. However, retrospective studies, including case reports/case series and qualitative studies, will be excluded. For mixed-method studies, if quantitative data can be extracted separately, the data will be included. No restrictions will be imposed on the publication language or publication status. (2) Types of participants: All types of HCWs, including physicians, nurses, hospital staff, and health managers, will be allowed. No restrictions were imposed on the participants’ sex, age, or ethnicity. However, even in studies published after the discovery of SARS-CoV-2 (i.e., December 2019), studies that did not describe whether participants were directly or indirectly affected by COVID-19 will be excluded. (3) Types of interventions: In this systematic review, mind–body modalities will include meditation, mindfulness-based intervention, autogenic training, yoga, tai chi, qigong, breathing exercise, music therapy, guided imagery, biofeedback, prayer, and faith-based techniques. (4) Types of controls: No treatment, wait-list, sham control, attention control, and active comparators will be allowed. (5) Types of outcome measures: The primary outcome was the level of perceived stress. Validated perceived stress evaluation tools such as the Perceived Stress Scale will be preferred [21], but a newly developed assessment tool reflecting the COVID-19 era will also be accepted. Secondary outcomes will include any mental health-related outcomes, such as depression, anxiety, and burnout. Safety data of the intervention used are also considered to be a secondary outcome of this review (Table 2).

### 2.3. Study Selection

In the first screening process, two independent researchers (C.-Y.K. and B.L.) will screen the titles and/or abstracts of the searched studies to identify potentially included articles. Second, potential reports will be sought for retrieval. Third, the same researchers will independently review the full texts of the retrieved reports. The final inclusion of the study will be decided through a three-step screening process. In the screening process, any disagreements between researchers will be resolved through discussion. We will evaluate the suitability of the literature according to the study design, population, and intervention eligibility criteria, and we will cite studies that appear to meet the inclusion criteria but are excluded and explain why they are excluded according to the PRISMA 2020 statement. The study selection process is presented in a PRISMA flow diagram (Figure 1).

### 2.4. Data Extraction

The following information will be extracted by two independent researchers (C.-Y.K. and B.L.) using a standardized, pre-defined, pilot-tested excel form: the first author’s name, year of publication, country, study design, sample size, details of participants, treatment and control interventions, duration of intervention, outcome measures, results, and safety data. Additionally, information for assessing the risk of bias (RoB) will be extracted. Discrepancies will be identified and resolved through discussion. When the data are insufficient, ambiguous, or missing, we will contact the corresponding authors of the original studies via e-mail. To prevent potential reporting bias by researchers, we will first perform data extraction on three studies by two researchers as a preliminary step and evaluate whether they agree. In addition, the suitability of data extraction will be evaluated by systematic review experts unrelated to the subject of this study.

### 2.5. Methodological Quality and Risk of Bias Assessment

Depending on the type of study, methodological quality will be assessed using the corresponding Critical Appraisal Skills Programme tools [22]. Additionally, to assess RoB, the Cochrane Collaboration RoB tool will be used in RCTs [23]. Using this tool, it is possible to evaluate RoB by classifying it into seven domains for the included RCTs. Accordingly, the methodological quality of included RCTs will be assessed in random sequence generation, allocation concealment, blinding of participants and personnel, blinding of outcome assessment, incomplete outcome data, selective reporting, and other domains with an assessment of “yes,” “no,” and “unclear.” In the case of other domains, if the demographic and clinical homogeneity between the treatment and control groups is presented at baseline, it will be evaluated as “low risk.” The Cochrane Handbook for Systematic Review of Interventions will guide the use of this tool [24]. The Quality Assessment of Controlled Intervention Studies by the National Heart, Lung, and Blood Institute (NHLBI) will be used for CCT [25]. The Quality Assessment Tool for Before-After (Pre-Post) Studies with no control group by NHLBI will be used for before-after studies [25]. Two independent researchers (C.-Y.K. and B.L.) will perform the methodological quality and RoB assessments of the included studies, and any disagreements between the researchers will be resolved through discussion. To prevent potential reporting bias by researchers, we will first perform a risk of bias assessment on three studies by two researchers as a preliminary step and evaluate whether they agree. In addition, the suitability of quality assessment will be evaluated by systematic review experts unrelated to the subjects of this study.

### 2.6. Data Analysis and Synthesis

Descriptive analysis will be used for all included studies. If sufficient homogeneous data from two or more RCTs exist, a quantitative synthesis (i.e., meta-analysis) will be performed using RevMan 5.4 (the Cochrane Collaboration, London, UK). Whether the data are sufficiently homogeneous will be judged based on the clinical homogeneity of the participants, interventions, controls, and outcomes in the included RCTs, assessed by two independent researchers (C.-Y.K. and B.L.), and resolved by consensus in case of disagreement. In the meta-analysis, dichotomous data will be presented as a risk ratio with 95% confidence intervals (CIs), and continuous data will be presented as mean differences with 95% CIs. Statistical heterogeneity between the studies in terms of effect measures will be assessed using both the χ^2^ test and the I^2^ statistic. The I^2^ values of ≥ 50% and ≥ 75% will be considered substantial and considerable heterogeneity, respectively. In the meta-analysis, a random-effects model will be used if the included studies have significant heterogeneity (I^2^ value ≥ 50%). However, the fixed-effect model will be used when the heterogeneity is not significant, or the number of studies included in the meta-analysis is small (less than five).

If the necessary data are available, subgroup analyses will be conducted according to the following criteria: (a) type of HCWs and (b) type of mind–body modality. When a sufficient number of studies exist in the meta-analysis, sensitivity analyses will be conducted to identify the robustness of the results by excluding (a) studies with high RoB and (b) data outliers.

### 2.7. Reporting Bias

If sufficient studies are available (i.e., more than 10 RCTs in each meta-analysis), evidence of publication bias will be evaluated visually by using funnel plots.

### 2.8. Patient and Public Involvement Statement

The present review protocol did not involve individual patients or public agencies.

## 3. Discussion

The COVID-19 outbreak poses an unprecedented challenge for HCWs, and their mental health concerns are in the spotlight [5]. In particular, nurses, female workers, front-line HCWs, younger medical staff, and workers in areas with higher infection rates have been reported to have mental health vulnerabilities in the COVID-19 era [26]. Maintaining the mental health and increasing the well-being of HCWs could be considered as a promising strategy to improve humanity’s capacity to cope with potential short- and long-term effects during and after the COVID-19 pandemic [5].

Mind–body modalities such as yoga, mindfulness-based intervention, and tai chi have been used to improve mental health and wellness in various populations, including HCWs [7,8,9,10]. In addition, mind–body modalities have recently been delivered in a non-face-to-face manner in a combined form with ICT, so mind–body modalities can be considered as a useful mental health strategy in the context of COVID-19, where social distancing is being emphasized [17].

However, for these mind–body modalities to be officially recommended for use in managing the mental health and well-being of HCWs in the COVID-19 era, their evidence base must be critically evaluated. For example, a UK research team suggested recommendations for guideline development for the mental health of HCWs using a qualitative study based on interviews with front-line HCWs in the UK and a systematic literature review methodology [27]. Importantly, as the team pointed out, guidelines for improving mental health and well-being in HCWs should provide evidence-based recommendations [27]. Portella et al., who analyzed relevant systematic reviews and controlled clinical studies, found that traditional, complementary, and integrative medicines, including mind–body modalities, could be useful, especially in the field of mental health, in the face of the COVID-19 pandemic [28]. In other words, despite the literature reporting the possibility of mind–body modalities being used in mental health problems related to COVID-19, the quantity and quality of the evidence in HCWs have not yet been comprehensively and critically evaluated. This, in turn, may hinder the consideration of mind–body modalities in developing mental health promotion guides for HCWs.

Therefore, this systematic review will comprehensively analyze the effectiveness of several mind–body modalities on the mental health of HCWs in the COVID-19 era. Although the primary outcome of this review is perceived stress, the outcomes to be considered will not be limited to this. The findings of this review will help establish public mental health strategies to improve the mental health and wellness of HCWs, which could potentially help humanity in the fight against the COVID-19 pandemic and improve the quality of care and conserve health resources.

## Figures and Tables

**Figure 1 healthcare-09-01320-f001:**
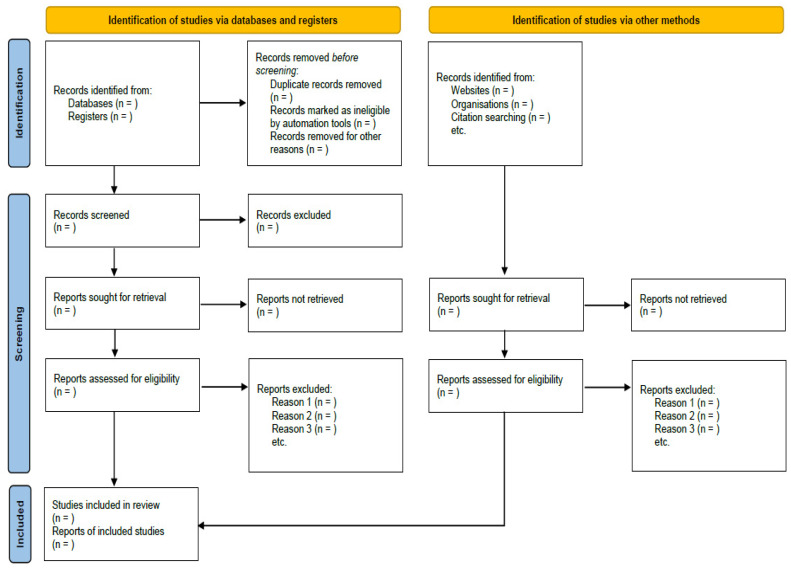
PRISMA flow diagram of the literature screening and selection process.

**Table 1 healthcare-09-01320-t001:** Search strategies for the Medline via PubMed.

#1. COVID-19[MH] OR SARS-CoV-2[MH] OR COVID-19[TIAB] OR SARS-CoV-2[TIAB] OR (wuhan[TIAB] AND coronavirus[TIAB]) OR 2019-nCoV[TIAB] OR 2019nCoV[TIAB] #2. Nurses[MH] OR Nursing[MH] OR nurs*[TIAB] #3. Physicians[MH] OR physician*[TIAB] OR doctor*[TIAB] #4. Health Personnel[MH] OR “health personnel*”[TIAB] OR “healthcare worker*”[TIAB] OR “hospital staff*”[TIAB] OR “health manager*”[TIAB] #5. “Mind–body Therapies”[MH] OR Meditation[MH] OR Mindfulness[MH] OR Relaxation[MH] OR “Relaxation Therapy”[MH] OR “Autogenic Training”[MH] OR Yoga[MH] OR “Tai Ji”[MH] OR Qigong[MH] OR “Breathing Exercises”[MH] OR “Music Therapy”[MH] OR “Imagery, Psychotherapy”[MH] OR “Biofeedback, Psychology”[MH] OR mind–body[TIAB] OR meditation[TIAB] OR mindful*[TIAB] OR relaxation[TIAB] OR “autogenic training”[TIAB] OR yoga[TIAB] OR “Tai Ji”[TIAB] OR “Tai Chi”[TIAB] OR Taichi[TIAB] OR qigong[TIAB] OR “qi gong”[TIAB] OR breathing[TIAB] OR music[TIAB] OR “guided imagery”[TIAB] OR biofeedback[TIAB] OR prayer[TIAB] OR faith[TIAB] #6. #1 AND (#2 OR #3 OR #4) AND #5

**Table 2 healthcare-09-01320-t002:** The inclusion criteria of this review in PICOS format.

Population	All types of HCWs, including physicians, nurses, hospital staff, and health managers
Intervention	Mind–body modalities, including meditation, mindfulness-based intervention, autogenic training, yoga, tai chi, qigong, breathing exercise, music therapy, guided imagery, biofeedback, prayer, and faith-based techniques
Comparison	No treatment, wait-list, sham control, attention control, and active comparators
Outcome	Primary outcome: level of perceived stressSecondary outcome: any mental health-related outcomes
Study type	All original clinical studies, including RCTs, CCTs, and before-after studies

Abbreviations. CCT, non-randomized controlled clinical trials; HCW, healthcare worker; RCT, randomized controlled trial.

## Data Availability

The data presented in this study are available in the article and Appendix A.

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
