# Peer review of "A Protocol for a Systematic Review and Meta-Analysis of Mind–Body Modalities to Manage the Mental Health of Healthcare Workers during the COVID-19 Era"

_healthcare, 2021, doi:10.3390/healthcare9101320_

Round 1

Reviewer 1 Report

Dear editor,

The submitted manuscript describes a protocol for a planned systematic review and/or meta-analysis synthesizing the literature in the area of mind-body (MB) modality interventions and mental health in health care workers. Given it follows the PRISMA guidelines of reporting, the structure and content of the protocol are very clear and detailed. I have provided a minor comments that may improve the manuscript:

  • The word ‘meanwhile’ (in abstract and 2nd paragraph of the introduction) indicates two events occurring at the same time. I think that MB interventions are more a solution to a problem (i.e., mental health because of COVID-19), not an event that occurs at the same time as the problem. Omitting the word ‘meanwhile’ and explicitly stating that MB interventions may provide a solution (i.e., may improve mental health of HCW during COVID-19) may be more fitting.
  • Line 67: ‘can be’ needs to be removed.
  • Line 68: There is a new PRISMA reporting guideline out (2020), please update and use the new one instead of the 2015 one.
  • The 2020 PRISMA guidelines also ask for additional information in the Flow chart, please update flowchart according to the 2020 guidelines.
  • Line 146: I am unsure how you would analyse the data extracted qualitatively, do you mean descriptive analyses? Descriptive analyses are quantitatively. If you are doing a qualitative analysis, please specify what method and what data you will be using to do so.
  • Line 158: you mention sufficient data, what do you consider sufficient data? Please specify the number of studies per group for the subgroup analysis.
  • Discussion: I feel it is merely a repetition of the introduction. Maybe have a stronger focus on what gaps this study will address and how the results can help inform quality of care and conserving health resources by linking how this has been done in the past.
  • I checked the OSF link and it did not work for me?

Overall, I think this is an interesting topic, which is important to investigate. I recommend to accept after these adjustments are made.

Author Response

  • Response to Comments from Reviewer 1

Overall comment:

Dear editor,

The submitted manuscript describes a protocol for a planned systematic review and/or meta-analysis synthesizing the literature in the area of mind-body (MB) modality interventions and mental health in health care workers. Given it follows the PRISMA guidelines of reporting, the structure and content of the protocol are very clear and detailed. I have provided a minor comments that may improve the manuscript:

Response:              

Thank you for the careful review of our manuscript.

Comment 1:

The word ‘meanwhile’ (in abstract and 2nd paragraph of the introduction) indicates two events occurring at the same time. I think that MB interventions are more a solution to a problem (i.e., mental health because of COVID-19), not an event that occurs at the same time as the problem. Omitting the word ‘meanwhile’ and explicitly stating that MB interventions may provide a solution (i.e., may improve mental health of HCW during COVID-19) may be more fitting.

Response 1:           

According to your advice, we omitted the word “meanwhile” and revised the sentences as recommended in the revised manuscript (Please see pages 1-2, marked in red).

“A wide range of mind-body modalities designed to facilitate the mind's capacity to affect health has been used for a long time to improve mental health, well-being, quality of life, and physical health of clinical populations, and the general population in various occupations [6]. Previous studies have reported that mind-body modalities such as yoga [7], mindfulness-based interventions [8,9], and tai chi [10] can benefit the health aspects of HCWs. Today, mind-body medicine has re-emerged for its mental health benefits in the context of COVID-19 [11,12], and some health systems use mind-body modalities to build strategies to improve the mental health of the public, patients with COVID-19, and HCWs [13,14].”

Comment 2:

Line 67: ‘can be’ needs to be removed.

Response 2:           

We have deleted these words, accordingly.

Comment 3:

Line 68: There is a new PRISMA reporting guideline out (2020), please update and use the new one instead of the 2015 one.

Response 3:           

Thank you for your comment. However, since this study is a protocol, we have reported according to the PRISMA-"Protocol" (the latest version is 2015). In the text, we have specified that future systematic review studies would be reported according to PRISMA 2020 (Please see page 2, marked in red).

“The protocol of this systematic review was registered in the Open Science Framework registry (https://osf.io/tudbw; registered on August 13, 2021). Any amendments to this systematic review will be tracked using this registry. This protocol was reported in accordance with the Preferred Reporting Items for Systematic Review and Meta-Analysis Protocols (PRISMA-P) 2015 statement [18]. We will report our systematic review according to the PRISMA 2020 statement [19].”

Comment 4:

The 2020 PRISMA guidelines also ask for additional information in the Flow chart, please update flowchart according to the 2020 guidelines.

Response 4:           

We updated Figure 1 according to PRISMA 2020 guidelines (Please see page 4).

Comment 5:

Line 146: I am unsure how you would analyse the data extracted qualitatively, do you mean descriptive analyses? Descriptive analyses are quantitatively. If you are doing a qualitative analysis, please specify what method and what data you will be using to do so.

Response 5:           

We will perform a descriptive analysis, and we also have modified the qualitative analysis to provide a descriptive analysis (Please see page 5, marked in red).

“Descriptive analysis will be used for all included studies.”

Comment 6:

Line 158: you mention sufficient data, what do you consider sufficient data? Please specify the number of studies per group for the subgroup analysis.

Response 6:           

We have modified our explanation of “sufficient data,” which was previously ambiguous, as follows (Please see page 5, marked in red).

“If sufficient homogeneous data from two or more RCTs exist, a quantitative synthesis (i.e., meta-analysis) will be performed using RevMan 5.4 (the Cochrane Collaboration, London, UK). Whether the data are sufficiently homogeneous will be judged based on the clinical homogeneity of the participants, interventions, controls, and outcomes in the included RCTs, assessed by two independent researchers (C.-Y.K. and B.L.), and resolved by consensus in case of disagreement. In the meta-analysis, dichotomous data will be presented as a risk ratio with 95% confidence intervals (CIs), and continuous data will be presented as mean differences with 95% CIs. Statistical heterogeneity between the studies in terms of effect measures will be assessed using both the χ² test and the I² statistic. The I² values of ≥ 50% and ≥ 75% will be considered substantial and considerable heterogeneity, respectively. In the meta-analysis, a random-effect model will be used if the included studies have significant heterogeneity (I² value ≥ 50%). However, the fixed-effect model will be used when the heterogeneity is not significant, or the number of studies included in the meta-analysis is small (less than five).”

Comment 7:

Discussion: I feel it is merely a repetition of the introduction. Maybe have a stronger focus on what gaps this study will address and how the results can help inform quality of care and conserving health resources by linking how this has been done in the past.

Response 7:           

We agree with your comments and have added information on the importance and potential impact of this systematic review to the Discussion (Please see page 6, marked in red).

“However, for these mind-body modalities to be officially recommended for use in managing the mental health and well-being of HCWs in the COVID-19 era, their evidence base must be critically evaluated. For example, the UK research team suggested recommendations for guideline development for the mental health of HCWs using a qualitative study based on interviews with front-line HCWs in the UK and a systematic literature review methodology [27]. Importantly, as the team pointed out, guidelines for improving mental health and well-being in HCWs should provide evidence-based recommendations [27]. Portella et al., who analyzed relevant 126 systematic reviews and controlled clinical studies, found that traditional, complementary, and integrative medicines, including mind-body modalities, could be useful, especially in the field of mental health, in the face of the COVID-19 pandemic [28]. In other words, despite literature reporting the possibility of mind-body modalities being used in mental health problems related to COVID-19, the quantity and quality of the evidence in HCWs have not yet been comprehensively and critically evaluated. This, in turn, may hinder the consideration of mind-body modalities in developing mental health promotion guides for HCWs.”

Comment 8:

I checked the OSF link and it did not work for me?

Response 8:           

Thank you for pointing this out. After checking the URL, we have revised the OSF link to the revised manuscript as follows: https://osf.io/tudbw.

Reviewer 2 Report

This is a study protocol to start a systematic literature review on the effect of mind-body therapies over stressed healthcare professionals. 
The design, presented with scholar correctness and methodological adequacy, seems adequate to reach the objectives of the research.
 The suggestions below are proposed to increase the value of the work. 
IMPORTANT: including or not any of them must be analyzed with the light of the study general scope.

A. In the research strategy, it would be interesting to include "prayer" and/or "faith-based techniques". They are widely used psycho-physiological resources, although they are not structured techniques like the others. 

B. During the manual search on Google Scholar for missing literature, will you check the journal reliability (adherent to peer-reviewed process)? If so, please state how at the manuscript; if not, state the reason to do so.
C. Would it be interesting to search for studies before the pandemia, but during catastrophes? On the one hand, healthcare professionals are also stressed at these occurrences; on the other hand, such stress is acute and self-limited. 

D. Instead begin in December 2019, why do the search period not start in January or February 2020? This is the time on which healthcare professionals started to be overloaded. Otherwise, please justify on the text the preference for December 2019.

E. A bias to be avoided is: did the subjects already practice any mind-body modality before the work overload? There are differences between people that started it now under extreme necessity and people practicing it for a long time. If this bias is inevitable, I suggest stating it on the section "Limitations" of the future publication.

F. Would it be interesting to include a broader concept of healthcare staff beyond direct patient care? They are important pieces in the gear of a hospital. I mean cleaning professionals, administrative employees, etc. In such a case, the included keyword would be "hospital staff", "health managers", etc.

Author Response

  • Response to Comments from Reviewer 2

Overall comment:

This is a study protocol to start a systematic literature review on the effect of mind-body therapies over stressed healthcare professionals.

The design, presented with scholar correctness and methodological adequacy, seems adequate to reach the objectives of the research.

The suggestions below are proposed to increase the value of the work.

IMPORTANT: including or not any of them must be analyzed with the light of the study general scope.

Response:              

Thank you for your careful review of the manuscript. We have made the needful changes.

Comment 1:

In the research strategy, it would be interesting to include "prayer" and/or "faith-based techniques". They are widely used psycho-physiological resources, although they are not structured techniques like the others.

Response 1:           

We have added prayer and faith-based techniques, as suggested. In addition, we have also revised the search strategy.

Comment 2:

During the manual search on Google Scholar for missing literature, will you check the journal reliability (adherent to peer-reviewed process)? If so, please state how at the manuscript; if not, state the reason to do so.

Response 2:           

Since our study will include gray literature and literature published in peer-reviewed journals, we will not evaluate the journal reliability of additional literature identified through manual searches in Google Scholar. However, we will indicate the source of the search for all included articles (see Page 2, marked in red) (Please see page 3, marked in red).

(3) Types of interventions: In this systematic review, mind-body modalities will include meditation, mindfulness-based intervention, autogenic training, yoga, tai chi, qigong, breathing exercise, music therapy, guided imagery, biofeedback, prayer, and faith-based techniques.

Comment 3:

  1. Would it be interesting to search for studies before the pandemia, but during catastrophes? On the one hand, healthcare professionals are also stressed at these occurrences; on the other hand, such stress is acute and self-limited.

Response 3:           

If we extend this period to catastrophes that occurred before COVID-19, we think the topic of this study will be different. Therefore, although we would like to limit the scope of this study to COVID-19, it would be appropriate to discuss the similarities and differences with existing studies while conducting a systematic review.

Comment 4:

Instead begin in December 2019, why do the search period not start in January or February 2020? This is the time on which healthcare professionals started to be overloaded. Otherwise, please justify on the text the preference for December 2019.

Response 4:           

We set the search start time to December 2019 because the first cases of COVID-19 were identified in December 2019. We have added relevant explanations and references to the revised manuscript based on your advice (Please see page 2, marked in red).

“One researcher (B.L.) will search the following six electronic bibliographic databases from December 2019 (when the first case of COVID-19 was identified [20]) to September 2021: …”

Comment 5:

A bias to be avoided is: did the subjects already practice any mind-body modality before the work overload? There are differences between people that started it now under extreme necessity and people practicing it for a long time. If this bias is inevitable, I suggest stating it on the section "Limitations" of the future publication.

Response 5:           

We believe that the experience of mind-body modality practice before work overload can be an important bias in interpreting the results. However, to comprehensively include as many relevant studies as possible, we intend to include literature without limitations. We will extract information from the included literature according to your valuable advice and add it to the study's limitations in future systematic reviews.

Comment 6:

Would it be interesting to include a broader concept of healthcare staff beyond direct patient care? They are important pieces in the gear of a hospital. I mean cleaning professionals, administrative employees, etc. In such a case, the included keyword would be "hospital staff", "health managers", etc.

Response 6:           

In response to this comment, we have revised our inclusion criteria and search strategies (Please see pages 2-3, marked in red).

Table 1. Search strategies for the Medline via PubMed

#1. COVID-19[MH] OR SARS-CoV-2[MH] OR COVID-19[TIAB] OR SARS-CoV-2[TIAB] OR (wuhan[TIAB] AND coronavirus[TIAB]) OR 2019-nCoV[TIAB] OR 2019nCoV[TIAB]

#2. Nurses[MH] OR Nursing[MH] OR nurs*[TIAB]

#3. Physicians[MH] OR physician*[TIAB] OR doctor*[TIAB]

#4. Health Personnel[MH] OR “health personnel*”[TIAB] OR “healthcare worker*”[TIAB] OR “hospital staff*”[TIAB] OR “health manager*”[TIAB]

#5. “Mind-Body Therapies”[MH] OR Meditation[MH] OR Mindfulness[MH] OR Relaxation[MH] OR “Relaxation Therapy”[MH] OR “Autogenic Training”[MH] OR Yoga[MH] OR “Tai Ji”[MH] OR Qigong[MH] OR “Breathing Exercises”[MH] OR “Music Therapy”[MH] OR “Imagery, Psychotherapy”[MH] OR “Biofeedback, Psychology”[MH] OR mind-body[TIAB] OR meditation[TIAB] OR mindful*[TIAB] OR relaxation[TIAB] OR “autogenic training”[TIAB] OR yoga[TIAB] OR “Tai Ji”[TIAB] OR “Tai Chi”[TIAB] OR Taichi[TIAB] OR qigong[TIAB] OR “qi gong”[TIAB] OR breathing[TIAB] OR music[TIAB] OR “guided imagery”[TIAB] OR biofeedback[TIAB] OR prayer[TIAB] OR faith[TIAB]

#6. #1 AND (#2 OR #3 OR #4) AND #5

“(2) Types of participants: All types of HCWs, including physicians, nurses, hospital staff, and health managers, will be allowed.”

Reviewer 3 Report

Dear Authors

Many thanks for submitting this manuscript. It provides a detail synopsis of what will be carried out in the actual systematic review which will have a potentially huge impact for the reader. However, some major edits are required for this paper to be published with the journal. 

Title of Manuscript

Suggest changing the title to state: "A Protocol for A Systematic Review and Meta-Analysis of Intervention Studies" 

Introduction

In line 30, I would suggest deleting the following "this coronavirus" as the ending of the sentence covers this detail. The final sentence of the first paragraph does not make sense when comparing it to the rest of the paragraph. My suggestion is to either delete or amend this sentence to allow it to flow with the remainder of the paragraph. You mention at the end of the second paragraph what mind-body modalities are, however, I feel this needs to be stated much earlier - and even in the abstract - so the reader can completely understand what they are about to read. In paragraph 3 you mention the examples of mind-body modalities again, this i feel was repetitive and did not work well with the flow of the manuscript. 

Material and Methods

You mentioned this protocol was registered with OSF, I would like to have seen the date it was registered also in the text. You also discuss the use of PRISMA 2015, however, PRISMA has since been updated and I feel you need to use the newer version of PRISMA in line with best evidence. This will involve the re-write of most of this manuscript but I feel this is important as PRISMA 2015 is now obsolete 

In table 1, there is no mention of search terms relating to COVID-19 which I think is a major error, considering the context of the article. In the eligability criteria, you do not make it clear whether you are focusing on just quantitative literature or expanding the search out to mixed method or qualitative literature. This is not identified until much later in the manuscript which is another major error that needs correcting in the revised manuscript. I also would have liked to have seen a table here showcasing the major points of the eligability criteria as as of now, the reader has to really examine the paper to understand what is excluded and what is not. 

In the study selection section, the screening process should involve three rounds of screening, not two. The second last sentence of this section does not make sense and should be either re-written or removed from the manuscript. 

Figure 1 comes from PRISMA 2015, there is now a new PRISMA flow diagram and this should be used here. 

In the following 2 sections - data extraction and risk of bias, you discuss risk of bias, but do not examine the reviewer's own potential risk of reporting bias which is needed here. Also within the risk of bias assessment, you discuss the assessment of quality - this needs to be a seperate section in the manuscript. Additionally, you should seriously consider using a recognised study quality assessment tool like CASP. 

Conclusion

As this is a protocol for a systematic review and meta-analysis, there is no need for a conclusion as one cannot be determined at this point. I would suggest amalgamating the conclusion text with the discussion text. 

General Overview

Although this text has much promise, much work needs to be done for this manuscript to be of publishable standard. I would also suggest to the authors to work on the use of the English language throughout the manuscript as currently, the manuscript is hard to read in parts.

Thank you for submitting the manuscript and I look forward to reading the revised submission.   

Author Response

  • Response to Comments from Reviewer 3

Overall comment:

Dear Authors

Many thanks for submitting this manuscript. It provides a detail synopsis of what will be carried out in the actual systematic review which will have a potentially huge impact for the reader. However, some major edits are required for this paper to be published with the journal.

Response:              

Thank you for the careful review of our manuscript. We have made the changes accordingly.

Comment 1:

Title of Manuscript

Suggest changing the title to state: "A Protocol for A Systematic Review and Meta-Analysis of Intervention Studies"

Response 1:           

We have changed the title of the manuscript as recommended.

Comment 2:

Introduction

In line 30, I would suggest deleting the following "this coronavirus" as the ending of the sentence covers this detail. The final sentence of the first paragraph does not make sense when comparing it to the rest of the paragraph. My suggestion is to either delete or amend this sentence to allow it to flow with the remainder of the paragraph. You mention at the end of the second paragraph what mind-body modalities are, however, I feel this needs to be stated much earlier - and even in the abstract - so the reader can completely understand what they are about to read. In paragraph 3 you mention the examples of mind-body modalities again, this i feel was repetitive and did not work well with the flow of the manuscript.

Response 2:           

We have revised both the Abstract and the Introduction based on your comments. Please check the red text in the revised manuscript (Please see page 1, marked in red).

“The coronavirus disease 2019 (COVID-19) pandemic has become an unprecedented threat to humanity worldwide, widely affecting not only physical health but also mental health [1]. In particular, healthcare workers (HCWs) have been exposed to the threat of severe acute respiratory syndrome coronavirus 2 (SARS-CoV-2) at the forefront since the early days of the pandemic [2,3]. In the early stage of the pandemic, lack of personal protective equipment (PPE), undiscovered etiology, strict isolation, high-intensity work, insufficient rest, fear of infection, and lack of a vaccine were major stressors for them [2,3]. Since then, some vaccines against SARS-CoV-2 have been developed and PPEs have been distributed, but their mental health is still threatened by a prolonged COVID-19 outbreak. According to a recent meta-analysis, HCWs exposed to stress from COVID-19 had significantly higher levels of anxiety (13.0 vs. 8.5%) and depression (12.2 vs. 9.5%) compared to experts in other fields [4]. Accordingly, the importance of improving the mental health of HCWs has emerged [5].”

Comment 3:

Material and Methods

You mentioned this protocol was registered with OSF, I would like to have seen the date it was registered also in the text. You also discuss the use of PRISMA 2015, however, PRISMA has since been updated and I feel you need to use the newer version of PRISMA in line with best evidence. This will involve the re-write of most of this manuscript but I feel this is important as PRISMA 2015 is now obsolete

Response 3:           

We have added the date of OSF registration to the revised manuscript. Additionally, since this study is a protocol, we have reported according to the PRISMA-"Protocol" (the latest version is 2015). We have specified in the text that future systematic reviews would be reported according to PRISMA 2020 (Please see page 2, marked in red).

“The protocol of this systematic review was registered in the Open Science Framework registry (https://osf.io/tudbw; registered on August 13, 2021). Any amendments to this systematic review will be tracked using this registry. This protocol was reported in accordance with the Preferred Reporting Items for Systematic Review and Meta-Analysis Protocols (PRISMA-P) 2015 statement [18]. We will report our systematic review according to the PRISMA 2020 statement [19].”

Comment 4:

In table 1, there is no mention of search terms relating to COVID-19 which I think is a major error, considering the context of the article. In the eligability criteria, you do not make it clear whether you are focusing on just quantitative literature or expanding the search out to mixed method or qualitative literature. This is not identified until much later in the manuscript which is another major error that needs correcting in the revised manuscript.

Response 4:           

Originally, we wanted to search as comprehensively as possible by not including search terms for COVID-19. However, I think your point is reasonable. Therefore, we have added a search term for COVID-19 in the revised manuscript (Please see page 3, marked in red).

Table 1. Search strategies for the Medline via PubMed

#1. COVID-19[MH] OR SARS-CoV-2[MH] OR COVID-19[TIAB] OR SARS-CoV-2[TIAB] OR (wuhan[TIAB] AND coronavirus[TIAB]) OR 2019-nCoV[TIAB] OR 2019nCoV[TIAB]

#2. Nurses[MH] OR Nursing[MH] OR nurs*[TIAB]

#3. Physicians[MH] OR physician*[TIAB] OR doctor*[TIAB]

#4. Health Personnel[MH] OR “health personnel*”[TIAB] OR “healthcare worker*”[TIAB] OR “hospital staff*”[TIAB] OR “health manager*”[TIAB]

#5. “Mind-Body Therapies”[MH] OR Meditation[MH] OR Mindfulness[MH] OR Relaxation[MH] OR “Relaxation Therapy”[MH] OR “Autogenic Training”[MH] OR Yoga[MH] OR “Tai Ji”[MH] OR Qigong[MH] OR “Breathing Exercises”[MH] OR “Music Therapy”[MH] OR “Imagery, Psychotherapy”[MH] OR “Biofeedback, Psychology”[MH] OR mind-body[TIAB] OR meditation[TIAB] OR mindful*[TIAB] OR relaxation[TIAB] OR “autogenic training”[TIAB] OR yoga[TIAB] OR “Tai Ji”[TIAB] OR “Tai Chi”[TIAB] OR Taichi[TIAB] OR qigong[TIAB] OR “qi gong”[TIAB] OR breathing[TIAB] OR music[TIAB] OR “guided imagery”[TIAB] OR biofeedback[TIAB] OR prayer[TIAB] OR faith[TIAB]

#6. #1 AND (#2 OR #3 OR #4) AND #5

We will only focus on quantitative intervention studies and mixed-method studies. If only quantitative data can be extracted separately, that data will be included. We have added this information to the revised manuscript (Please see page 3, marked in red).

“The inclusion criteria for this systematic review are as follows: (1) Types of study: Although randomized controlled clinical trials (RCTs) are a type of clinical study with the highest level of evidence to evaluate the effectiveness and efficacy of the intervention, given the urgency of COVID-19, meaningful results can be found not only in RCTs but also in other intervention studies besides RCT. Therefore, this systematic review will allow for all types of quantitative intervention studies. Therefore, all original clinical studies, including RCTs, non-randomized controlled clinical trials (CCTs), and before-after studies will be conducted. However, retrospective studies, including case reports/case series and qualitative studies will be excluded. For mixed-method studies, if quantitative data can be extracted separately, the data will be included. No restrictions will be imposed on the publication language or publication status.”

Comment 5:

I also would have liked to have seen a table here showcasing the major points of the eligability criteria as as of now, the reader has to really examine the paper to understand what is excluded and what is not.

Response 5:           

In future systematic reviews, we will evaluate the suitability of the literature according to the study design, population, and intervention eligibility criteria, as well as studies that appear to meet the inclusion criteria, but which are excluded, and explain why they are excluded according to the PRISMA 2020 statement. We have added this information to the revised manuscript (Please see page 3, marked in red).

“We will evaluate the suitability of the literature according to the study design, population, and intervention eligibility criteria, and we will cite studies that appear to meet the inclusion criteria but are excluded and explain why they are excluded according to the PRISMA 2020 statement.”

Comment 6:

In the study selection section, the screening process should involve three rounds of screening, not two. The second last sentence of this section does not make sense and should be either re-written or removed from the manuscript.

Response 6:           

We have revised the study selection section according to the PRISMA 2020 statement and removed the second last sentence of this section (Please see page 3, marked in red).

“In the first screening process, two independent researchers (C.-Y.K. and B.L.) will screen the titles and/or abstracts of the searched studies to identify potentially included articles. Second, potential reports will be sought for retrieval. Third, the same researchers will independently review the full texts of the retrieved reports. The final inclusion of the study will be decided through a three-step screening process.”

Comment 7:

Figure 1 comes from PRISMA 2015, there is now a new PRISMA flow diagram and this should be used here.

Response 7:           

We have updated Figure 1 according to PRISMA 2020 guidelines.

Comment 8:

In the following 2 sections - data extraction and risk of bias, you discuss risk of bias, but do not examine the reviewer's own potential risk of reporting bias which is needed here. Also within the risk of bias assessment, you discuss the assessment of quality - this needs to be a seperate section in the manuscript. Additionally, you should seriously consider using a recognised study quality assessment tool like CASP.

Response 8:           

To prevent potential reporting bias by researchers, we will first perform data extraction and risk of bias assessment on three studies by two researchers as a preliminary, and evaluate whether they agree. In addition, the suitability of data extraction and risk of bias assessment will be evaluated by systematic review experts unrelated to the subject of this study. Additionally, we will evaluate the methodological quality of our future systematic review using the AMSTAR 2 tool, a critical appraisal tool for systematic reviews, and present the evaluation results in an appendix. We have added this information to the revised manuscript (Please see page 5, marked in red).

“To prevent potential reporting bias by researchers, we will first perform a risk of bias assessment on three studies by two researchers as a preliminary step and evaluate whether they agree. In addition, the suitability of quality assessment will be evaluated by systematic review experts unrelated to the subjects of this study.”

2.9. Quality assessment of systematic review

We will evaluate the methodological quality of our future systematic review with the assessment of multiple systematic reviews 2 tool (AMSTAR-2) [25], a critical appraisal tool for systematic reviews, and present the evaluation results as an appendix.”

Comment 9:

Conclusion

As this is a protocol for a systematic review and meta-analysis, there is no need for a conclusion, as one cannot be determined at this point. I would suggest amalgamating the conclusion text with the discussion text.

Response 9:           

We have added the Conclusion within the Discussion, according to your advice,

Comment 10:

General Overview

Although this text has much promise, much work needs to be done for this manuscript to be of publishable standard. I would also suggest to the authors to work on the use of the English language throughout the manuscript as currently, the manuscript is hard to read in parts.

Response 10:         

Our manuscript has been proofread by an English proofreading company, and we have attached a relevant certificate.

Reviewer 4 Report

I have a problem with the evaluation of the text. I usually evaluate texts where there are test results. And here we only have information about the literature review. In my opinion, such information has no value. The text written on the basis of this review will be able to be assessed. 

Author Response

  • Response to Comments from Reviewer 4

Overall comment:

I have a problem with the evaluation of the text. I usually evaluate texts where there are test results. And here we only have information about the literature review. In my opinion, such information has no value. The text written on the basis of this review will be able to be assessed.

Response:              

Our manuscript is a protocol for a systematic review and meta-analysis of mind-body modalities to manage healthcare workers' mental health during the COVID-19 era. We would like to publish this protocol to prevent potential reporting bias in the systematic review process, and this protocol has been reported in strict accordance with PRISMA-P. In addition, we have modified the paper to improve the methodological quality of this study, according to the opinions of the three other reviewers. Once again, we ask for a careful review of this manuscript, and please let us know if there are any other relevant corrections to be made.

Round 2

Reviewer 3 Report

Dear authors

Many thanks for submitting the reviewed manuscript and for taking our points on board. I only have three small edits that I feel are required before publication.

Firstly, in section 2.2, you describe the eligibility criteria in word format. This is valuable for the reader. However, I believe making the points extra clear via an inclusion'/exclusion criteria table would support the reader in fully understanding what it is you are trying to achieve. 

Secondly in the red text of the discussion, on line 210, you state "who analyzed relevant 126 systematic reviews..." Please pick either relevant or 126 and edit the sentence to allow this point to make sense. 

Lastly, in section 2.9 - quality assessment, you state the use of AMSTAR-2. This is only for systematic reviews and not for the data you will be reviewing. Please change this to another recognized quality appraisal tool for quantitative research. 

Thank you again for submitting the manuscript.  

Author Response

  • Response to Comments from Reviewer 3

Overall comment:

Dear authors

Many thanks for submitting the reviewed manuscript and for taking our points on board. I only have three small edits that I feel are required before publication.

Response:              

Thank you for the careful review of our manuscript.

Comment 1:

Firstly, in section 2.2, you describe the eligibility criteria in word format. This is valuable for the reader. However, I believe making the points extra clear via an inclusion'/exclusion criteria table would support the reader in fully understanding what it is you are trying to achieve.

Response 1:           

According to your advice, we have added Table 2, which describes the eligibility criteria in PICOS format. We hope this will increase the readability of our readers. (Please see page 3, marked in red)

Comment 2:

Secondly in the red text of the discussion, on line 210, you state "who analyzed relevant 126 systematic reviews..." Please pick either relevant or 126 and edit the sentence to allow this point to make sense.

Response 2:           

Thank you. We have deleted 126. (Please see page 6)

“Portella et al., who analyzed relevant systematic reviews and controlled clinical studies, found that traditional, complementary, and integrative medicines, including mind-body modalities, could be useful, especially in the field of mental health, in the face of the COVID-19 pandemic [28].”

Comment 3:

Lastly, in section 2.9 - quality assessment, you state the use of AMSTAR-2. This is only for systematic reviews and not for the data you will be reviewing. Please change this to another recognized quality appraisal tool for quantitative research.

Response 3:           

Thank you for your comment. We removed the descriptions of AMSTAR-2. Instead we added CASP you recommended earlier. (Please see pages 3-4)

“2.5. Methodological quality and risk of bias assessment

Depending on the type of study, methodological quality will be assessed using the corresponding Critical Appraisal Skills Programme tools [22]. Also to assess RoB, the Cochrane Collaboration RoB tool will be used in RCTs [23]. Using this tool, it is possible to evaluate RoB by classifying it into seven domains for the included RCTs. Accordingly, the methodological quality of included RCTs will be assessed in random sequence generation, allocation concealment, blinding of participants and personnel, blinding of outcome assessment, incomplete outcome data, selective reporting, and other domains with an assessment of “yes,” “no,” and “unclear.” In the case of other domains, if the demographic and clinical homogeneity between the treatment and control groups is presented at baseline, it will be evaluated as “low risk.” The Cochrane Handbook for Systematic Review of Interventions will guide the use of this tool [24]. The Quality Assessment of Controlled Intervention Studies by the National Heart, Lung, and Blood Institute (NHLBI) will be used for CCT [25]. The Quality Assessment Tool for Before-After (Pre-Post) Studies with no control group by NHLBI will be used for before-after studies [25]. Two independent researchers (C.-Y.K. and B.L.) will perform the methodological quality and RoB assessments of the included studies, and any disagreements between the researchers will be resolved through discussion. To prevent potential reporting bias by researchers, we will first perform a risk of bias assessment on three studies by two researchers as a preliminary step and evaluate whether they agree. In addition, the suitability of quality assessment will be evaluated by systematic review experts unrelated to the subjects of this study.”

Reviewer 4 Report

There are no objections to the text. It meets the publication standards. 

Author Response

Thank you for your careful review of the manuscript.